# The post-activation performance enhancement effect of different plyometric training modalities on short-distance sprinting: An acute randomized crossover study

Wenhao Qu[1,2], Wuwen Peng[1,2], Yueming Li[1,3], Lin Xie[2,3,4], Jian Sun[1,2,3]*, Duanying Li[1,2,3]*

**1** School of Athletic Training, Guangzhou Sport University, Guangzhou, Guangdong, China, **2** Guangdong Provincial Key Laboratory of Human Sports Performance Science, Guangzhou, Guangdong, China, **3** Key Laboratory of Human-Computer Intelligent Interaction for Athletic Performance and Health Promotion, Guangzhou Sport University, Guangzhou, Guangdong, China, **4** Macao Polytechnic University, Macao, China

☯ These authors contributed equally to this work
* liduany@gzsport.edu.cn; sunjian@gzsport.edu.cn

## Abstract

### Objective

This study aimed to investigate the effects of different plyometric training modalities [vertical jump plyometric training (VJ-PT), horizontal jump plyometric training (HJ-PT), and combined jump plyometric training (CJ-PT)] on post-activation performance enhancement (PAPE) in short-distance sprint performance.

### Methods

A randomized crossover design was employed, with 12 participants (sex: male; age:19.6 ± 0.9; BMI:24.9 ± 3.85). recruited for this study. Participants underwent three training interventions: VJ-PT, HJ-PT, and CJ-PT. Each training protocol consisted of 2 sets × 6 repetitions of one of the jump training modalities. The Smart Speed system was used to assess 5-meter sprint performance pre-intervention and at 4, 8, 12, and 16 minutes post-training. Repeated-measures ANOVA and Pearson correlation analyses were conducted using SPSS 26.0 and JASP 18.3.

### Results

No significant effects were observed for time (F = 1.43, p = 0.23), intervention (F = 0.32, p = 0.72), or interaction (F = 1.03, p = 0.41). However, VJ-PT demonstrated moderate effect sizes for 5-meter sprint performance across post-training time points, with larger effects observed at 4–8 minutes. HJ-PT and CJ-PT exhibited small negative effects, with no significant PAPE effects detected. Furthermore, Pearson correlation analysis indicated no significant associations between sprint performance and the time to peak PAPE after any exercise (P > 0.05).

**Data availability statement:** All relevant data are within the manuscript and its Supporting Information files.

**Funding:** The Guangdong Provincial Philosophy and Social Sciences Regularization Project 2022 (GD22CTY09) had no role in study design, data collection or analysis, the decision to publish, or the preparation of the manuscript.

**Competing interests:** The authors have declared that no competing interests exist.

## Conclusion

This study provides preliminary insights into the short-term effects of different plyometric-based conditioning activities (CAs) on short-distance sprint performance. Vertical jump plyometric training showed potential benefits, though findings are limited by small sample size and no control group. Horizontal and combined training did not produce significant PAPE effects, likely due to differences in time-to-peak and cumulative fatigue. Future studies should include a larger sample size, further investigate responses in both sexes, control for confounding factors, and use surface electromyography to clarify the interactions between CA types and recovery.

## Introduction

The neuromuscular system may experience fatigue or potentiation in response to voluntary or involuntary stimuli, thereby regulating various human life activities such as walking, running, and jumping [1,2]. Post-activation potentiation (PAP) is a common physiological phenomenon whereby the neuromuscular system performance is enhanced through maximal or near-maximal voluntary contractions, resulting in an acute increase in force production and in the rate of force development. [3,4]. This potentiation arises from the phosphorylation of myosin regulatory light chains (MRLC) induced by a single intense stimulation, which increases the calcium sensitivity of the myosin complex, increase the rate of cross-bridge formation, and elicits an acute increase in skeletal muscle contractile force [5–7]. Over the years, this principle has become a classic component of sports training and has been increasingly translated into practical applications within sport performance settings. [8]. However, some studies have indicated that athletic performance does not necessarily improve following PAP induction [7], and performance may even be enhanced once the PAP effect has subsided (>3min) [9–11]. Increasing evidence indicates that the peak of enhanced voluntary muscle contraction strength typically occurs 6–10 minutes after the conditioning activity [7,12], which cannot be satisfactorily explained by the PAP effect alone. Furthermore, studies have found that the potentiation of muscle twitch force (torque) induced by PAP does not affect or has minimal impact on athletic performance after a conditioning activity (CA) [11]. To address this, Cuenca-Fernández and colleagues proposed the concept of post-activation performance enhancement (PAPE) [12] to explain the acute improvements in athletic performance. However, PAPE is not entirely independent of PAP and may be related to residual PAP effects in the initial phase following the conditioning intervention or to other factors such as muscle temperature [13], fiber water content [14–16], and increased recruitment of high-threshold motor units [7,17].

A review of the literature has shown that various conditioning activities (CA) can induce PAPE effects to varying degrees, thereby positively influencing athletic performance [18–25], for example: bench press (BP), barbell squat (BS), complex training (CT), flywheel resistance training(FRT), and Plyometric Training (PT). Recent meta-analytic data indicate that Plyometric Training (ES = 0.42) induces a greater

amplitude of PAPE compared to traditional resistance exercise (ES = 0.23), maximal isometric voluntary contraction (ES = 0.31), and other CA types (ES = 0.24), suggesting it may be the optimal CA for eliciting PAPE [26]. PT is an effective physical training method that utilizes the stretch-shortening cycle (SSC) mechanism [27]. Simultaneously, several studies have demonstrated that Plyometric training recruits more high-threshold motor units than traditional resistance training, a neuromuscular activation pattern that provides the physiological basis for enhanced explosive power [28,29]. In addition, Plyometric training simulates the specific movement patterns in competitions (such as jumping and changing direction), with low technical demands, enabling athletes to focus more effectively on competitive requirements. Therefore, PT may be an ideal pre-competition activation strategy [30–32]. However, existing evidence predominantly focuses on comparative studies between potentiated training and other CA types. Few studies have investigated the relative advantages of different Plyometric protocols for inducing PAPE. Similarly, studies indicate that research on the PAPE effects of potentiated protocols for activities requiring vertical force vectors (e.g., jumping) predominates, whereas investigations into horizontal force vectors (e.g., sprinting) and optimal rest intervals are limited and lack a clear consensus [33,34].

It is noteworthy that although high-intensity CA can elicit more pronounced potentiation effects during PAPE induction, they also exacerbate fatigue accumulation, which may diminish the net benefit of the enhancement [5]. Therefore, to reduce fatigue accumulation, this study employs a low ground contact frequency (LGCF) Plyometric paradigm [5] to investigate differences in PAPE regulation on sprint performance among three potentiated training protocols, with the aim of providing strategies for enhanced physical activation before competition or training.

## Materials & methods

### Study design

This study employed a randomized crossover design to investigate the acute effects of vertical jump plyometric training, horizontal jump plyometric training, and combined jump plyometric training on short-distance sprint performance. The recruitment period for participants in this study was from May 15th to 27th, 2025.Prior to commencing the experiment, a Latin square design was used to arrange the sequence of interventions. All participants completed two familiarization sessions to ensure they were fully comfortable with the sprint tests and the various plyometric exercises before the experimental sessions. During the experimental phase, subjects performed the three intervention protocols in randomized order, with a minimum 72 hour interval between each protocol to eliminate residual effects and fatigue. In accordance with [5], sprint performance was reassessed at 4 min, 8 min, 12 min, and 16 min after each training session (Fig 1). All testing and training sessions were conducted at the same facility under the direct supervision of two researchers.

### Participants

Based on published findings on PAPE's effects on jump performance with α = 0.05 and 1−β = 0.95 (Seitz & Haff, 2016), t the sample size was determined to be 12 participants using G*Power software (Version 3.1.9.7). Based on prior participant classification research (McKay et al., 2022), detailed inclusion criteria were established:(1) no history of neuromuscular disorders; (2) no lower-limb injury in the six months preceding the study; (3) Complete 3–5 weekly sessions that are consistently scheduled and primarily emphasize strength and speed training. This study was conducted in accordance with the Declaration of Helsinki, and participants signed written informed consent after being fully informed of the associated risks and benefits. This study was approved by the Ethics Committee of the Guangzhou Sport University (approval No. 2025LCLL-061). The study received approval from the Chinese Clinical Trial Center (approval no. ChiCTR2500103142). More detailed information about the participants is provided in Table 1 below.

Plyometric Conditioning Activity Protocols Before the experiment, measures were taken to prevent the potential confounding effect of elevated muscle temperature on the accuracy of PAPE test results [35]. All participants completed an efficient standardized warm-up lasting 5–10 minutes. The specific warm-up protocol comprised dynamic muscle stretches,

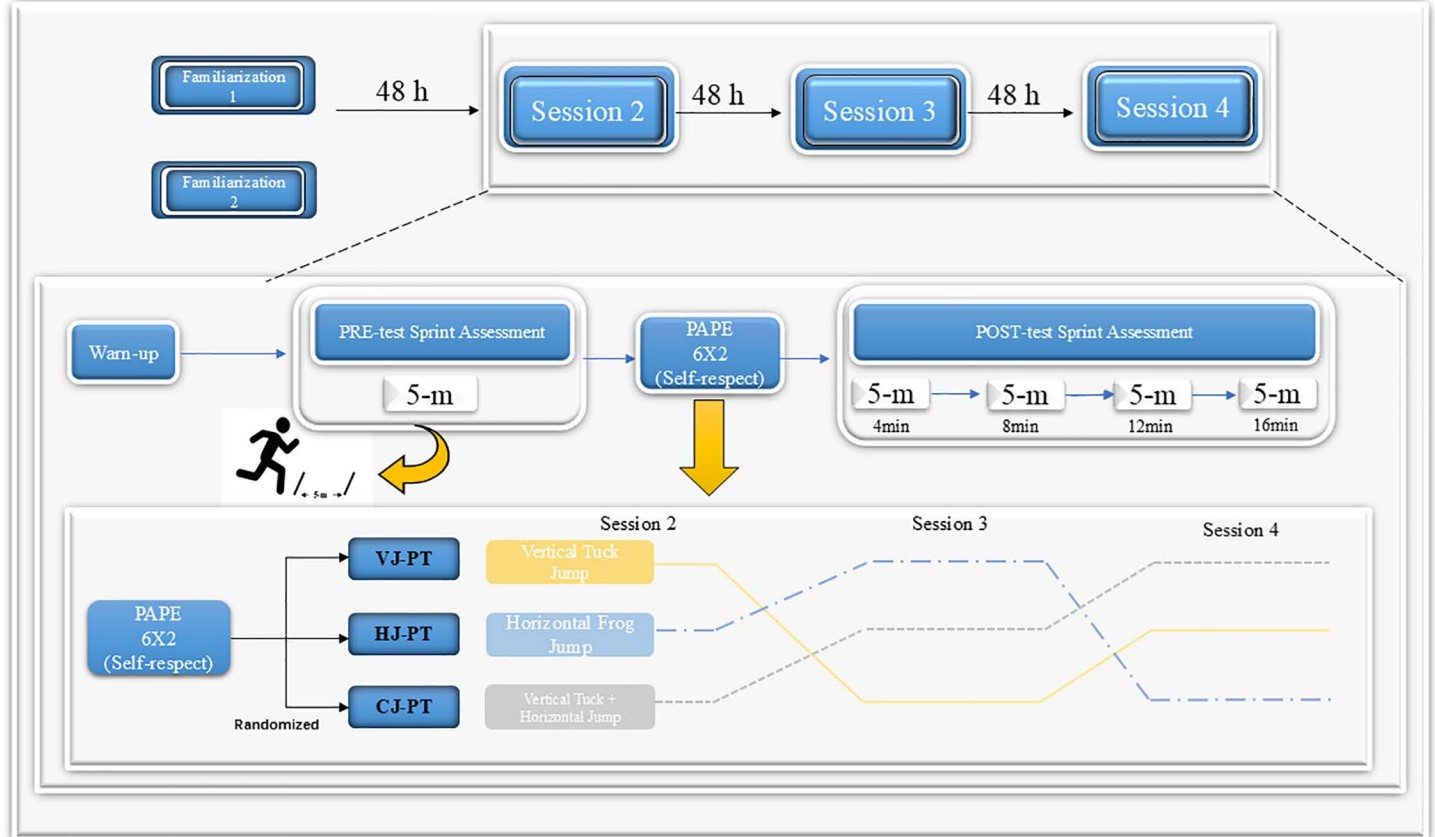

**Fig 1. Experimental flow chart.**

**Table 1. Basic Information of the Subjects (n = 12).**

| index | M±SD |
|---|---|
| Height (cm) | 173.4 ± 3.37 |
| Weight (kg) | 75.2 ± 13.27 |
| Age (years) | 19.6 ± 0.9 |

including cradle-leg pulls, Y-stretches, the world's greatest stretch, inchworm crawls, and bear crawls. Moreover, to minimize residual PAPE effects following the pre-test [36], The activation intervention was performed 15 min after the pre-test, and sprint performance was assessed at 4, 8, 12, and 16 min following each training session [12,37–39], with two researchers supervising and providing verbal encouragement.

The HJ-PT group performed the Horizontal Frog Jump, the VJ-PT group performed the Vertical Tuck Jump, and the CJ-PT group performed a Vertical Tuck Jump immediately followed by a Horizontal Frog Jump; the exercises and loading parameters for the three Plyometric training modalities are presented in Table 2. According to the guidelines of the National Strength and Conditioning Association, the protocol for the Vertical Tuck Jump involves standing with feet shoulder-width apart, descending to a 90° knee flexion, explosively jumping upward while tucking the knees toward the chest until they touch the torso, and upon landing immediately initiating the next jump. The Horizontal Frog Jump is

**Table 2. Information on Intervention Methods.**

| Conditioning Activity | Training movements | Sets | Repetition | Load | Inter -set interval |
|---|---|---|---|---|---|
| VJ-PT | Vertical Tuck Jump | 2 | 6 | Bodyweight | 30s |
| HJ-PT | Horizontal Frog Jump | | | | |
| CJ-PT | Vertical Tuck Jump+ Horizontal Frog Jump | | | | |

executed by standing with feet slightly wider than shoulder-width, performing an arm swing while squatting to the deepest position, then forcefully driving both legs to propel the body forward and upward, achieving full extension in the air. Upon landing, both feet contact the ground first, followed by a natural hip and knee flexion to absorb the impact, after which the next jump is immediately initiated.

### Linear sprint 5 m

Research indicates that shorter sprint distances exhibit a stronger correlation with peak power output [40]; therefore, this study employed a 5-m sprint to capture the PAPE effect.. Sprint performance over 5-m was recorded using the Smart Speed system (VALD Performance, Brisbane, Australia.). Both the start and finish lines comprised two sensor units each, positioned approximately 3-m apart. Throughout the trials, researchers provided continuous verbal encouragement to participants. Participants assumed a three-point stance behind the start line, commenced the sprint at their own volition, and accelerated maximally toward the finish, with times recorded to 0.01s.

### Statistical analysis

Descriptive statistics are presented as mean ± SD; significance was set at $p < 0.05$. Data were analyzed using SPSS26.0 and JASP18.3; normality and homogeneity of variance were assessed via Shapiro–Wilk and Levene's tests. A 3 × 5 repeated-measures Analysis of Variance (RM-ANOVA) was used to examine differences among the three potentiation protocols, time points (4, 8, 12, 16 min), and their interaction. The ANOVA effect sizes (ES) were reported as $\eta_p^2$, and within- intervention ES as Cohen's d [41] was reported. $\eta_p^2$ values of 0.01–0.059, 0.059–0.138, and >0.138 indicate small, medium, and large ES, respectively; Cohen's d thresholds of 0.2–0.5, 0.5–0.8, and >0.8 denote small, medium, and large ES [42]. Pearson correlation analysis was used to assess the relationship between sprint performance and time to peak PAPE. Qualitative descriptors for correlation magnitude were: <0.10 negligible, 0.10–0.29 small, 0.30–0.49 moderate, 0.50–0.69 large, 0.70–0.89 very large, 0.90–0.99 nearly perfect, and 1.0 perfect [43].

## Results

As shown in Table 3, the main effect of time (F = 1.43, p = 0.23, $\eta_p^2$=0.04), the main effect of intervention (F = 0.32, p = 0.72, $\eta_p^2$=0.02), and the interaction effect (F = 1.03, p = 0.41, $\eta_p^2$=0.05) were all non-significant and exhibited small effect sizes (Fig 2). VJ-PT exhibited medium effect sizes for 5 m acceleration at all post-intervention time points; HJ-PT showed small to moderate detrimental effects; and CJ-PT presented small effect sizes (Table 3). Furthermore, the between-intervention effect sizes following PAPE are depicted in Fig 3. Moreover, Pearson's correlation analysis revealed no significant relationship between sprint performance and the time to peak PAPE induced by any of the protocols (VJ-PT: p = 0.47, r = 0.62; HJ-PT: p = 0.91, r = 0.16; MJ-PT: p = 0.87, r = 0.23).

## Discussion

This investigation employed a randomized parallel crossover design to examine the acute effects of Vertical Jump Plyometric Training (VJ-PT), Horizontal Jump Plyometric Training (HJ-PT), and Combined jump plyometric training (CJ-PT) on 5 m sprint

**Table 3. Time Course of Post-Activation Performance Enhancement for 5m Sprinting Following Three Different Activation Methods.**

| Conditioning Activity | Baseline | Time points | | | | | | | | Group | | Group×time | |
|---|---|---|---|---|---|---|---|---|---|---|---|---|---|
| | | 4min | | 8min | | 12min | | 16min | | | | | |
| | | Result | Cohen's d | Result | Cohen's d | Result | Cohen's d | Result | Cohen's d | p | $\eta_p^2$ | p | $\eta_p^2$ |
| VJ-PT | 1.20±0.10 | 1.18±0.07 | 0.37 | 1.15±0.10 | 0.57 | 1.16±0.08 | 0.40 | 1.18±0.10 | 0.20 | 0.44 | 0.03 | 0.56 | 0.05 |
| HJ-PT | 1.16±0.07 | 1.17±0.09 | 0.11 | 1.16±0.08 | $9.2\times10^{-4}$ | 1.18±0.09 | −0.30 | 1.16±0.09 | −0.05 | | | | |
| CJ-PT | 1.24±0.11 | 1.20±0.08 | 0.48 | 1.22±0.07 | 0.28 | 1.20±0.10 | 0.42 | 1.20±0.06 | 0.37 | | | | |

Note: VJ-PT = Vertical jump-Plyometric training; HJ-PT = Horizontal jump-Plyometric training; CJ-PT = Combined jump-Plyometric training.

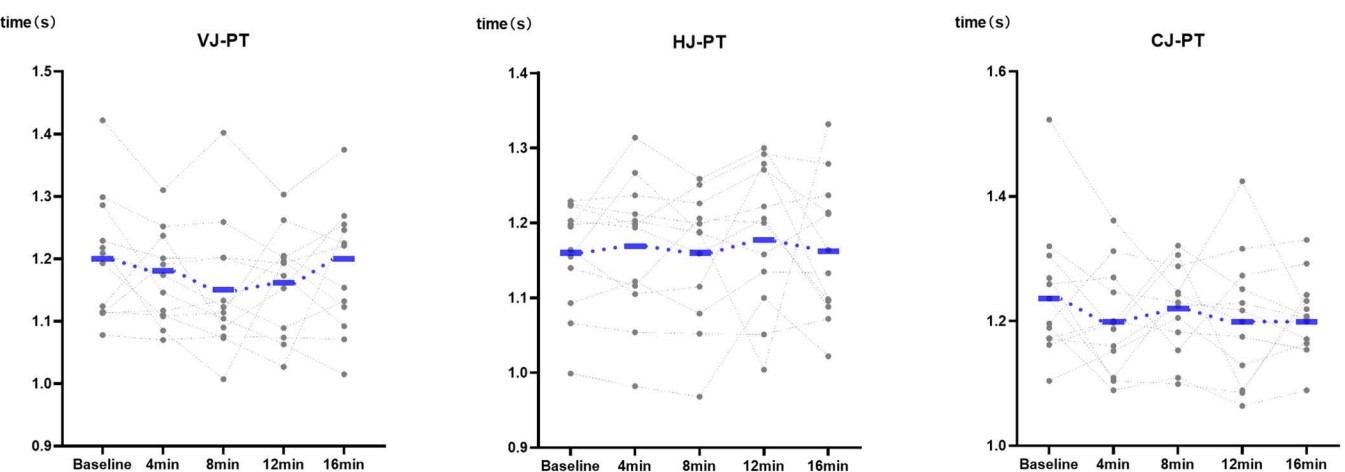

**Fig 2. Effects of Plyometric Training Modalities on SLJ and 5-m sprint performance and individual participant results. Intervention.**

performance. Experimental findings suggest that VJ-PT demonstrated a non-significant moderate effect of preference and reached a potential peak PAPE advantage during minutes 4–8, yet the lack of a no-intervention control group in this study warrants cautious interpretation of these trends. HJ-PT and CJ-PT did not exhibit potential PAPE benefits, and their efficacy remains to be confirmed.

This result is supported by previous research, which found that in female futsal players, sprint training and vertical plyometric training did not differ significantly in eliciting the PAPE effect [44]. Compared with horizontal jumping, the research has found that the vertical-directional enhanced movement has a higher recruitment of muscle fibers [45]. Therefore, we speculate that VJ-PT can generate sufficient potentiation to improve sprint performance. In short-distance acceleration, net horizontal force determines the magnitude of acceleration [46–49]. Previous research indicates that elite sprinters generate greater horizontal impulse during the acceleration phase, resulting in higher acceleration [50]. Consequently, this could account for the absence of immediate improvements following VJ-PT. Similarly, training protocols that exhibit a higher degree of similarity to the movement patterns during exercise have been shown to produce greater transfer effects in terms of explosive power improvements compared to non-specific movement patterns. [47]. However, during sprinting, mechanical variables are entangled, and no single variable is associated with better performance [51] This may account for the absence of an immediate sprint performance benefit following HJ-PT. Moreover, horizontal-jump drills typically involve ground-contact times exceeding 100ms [52], which are markedly longer than the stance phase of sprinting (~80–90ms) [51]; this discrepancy may further magnify measurement bias in the PAPE response.

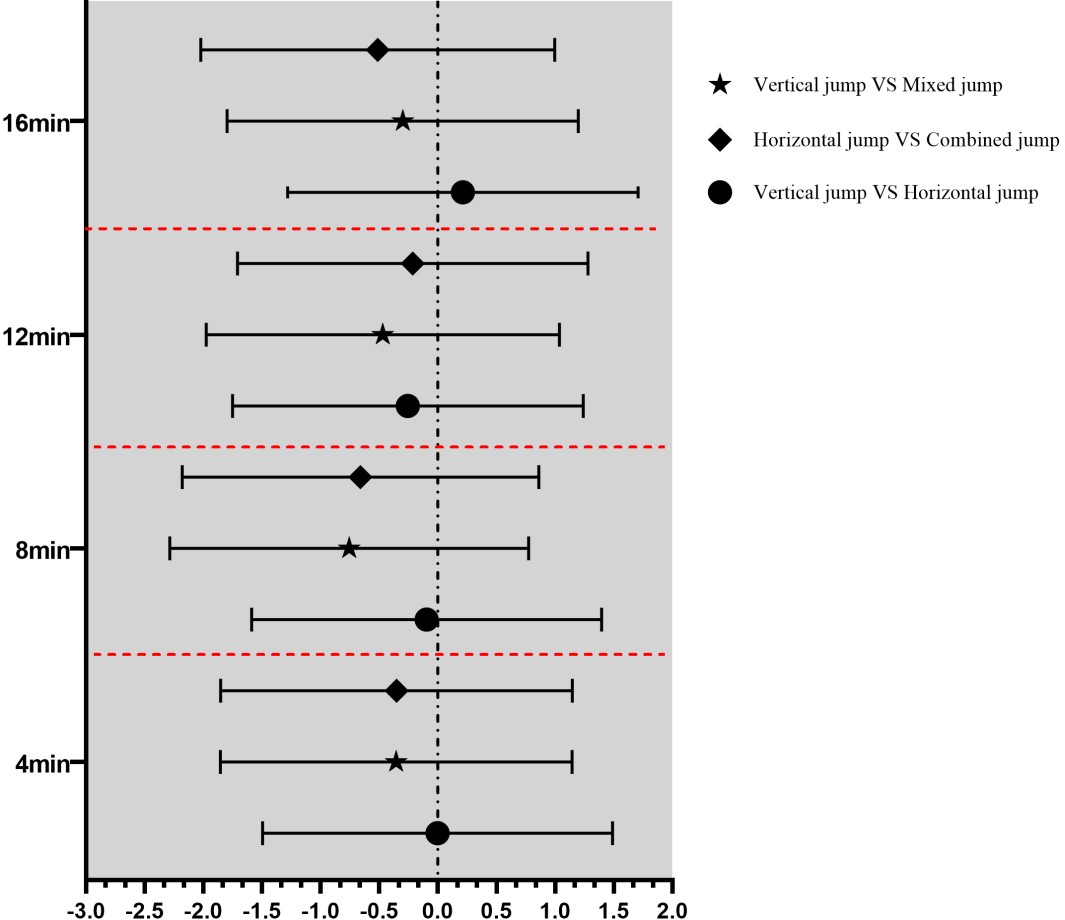

**Fig 3. Effect Size (Cohen's d) for Intervention (Post-Activation Plyometric Protocol) Comparisons.**

It is noteworthy that individual athletes may differ in their reliance on horizontal versus vertical force components during sprinting. Some athletes may rely more on increased step frequency to boost speed, whereas others may emphasize step length, which can confer greater benefits [53,54]. Consequently, Plyometric training movements should closely replicate sport-specific actions to maximally engage analogous neuromuscular reflex pathways. Although the jump training parameters for all three CA were kept constant, the Combined jump protocol yielded poorer sprint results. We hypothesize that this may be because its more complex movement sequence demands additional neuromuscular control and leads to greater fatigue accumulation compared to the Vertical and Horizontal Jump interventions. Hence, when employing potentiation as an enhancement strategy, considerations should include the direction of force production and the similarity of the movement pattern to the specific sporting activity.

This study also investigated the temporal PAPE response in relation to 5 m sprint performance; although no significant differences were detected among the three CA, within-intervention effect sizes indicated that vertical intervention exhibited its greatest effect size at the 8minute post-activation test, reaching a moderate magnitude. However, these trend-like findings should be interpreted with caution. Piper A.D. and Turner et al. found that performance enhancement after VJ-PT was greater following a longer rest period (8 min) than after a shorter one (4 min) [55]. However, However, other studies have found that Till and Cooke observed no improvement in 10m and 20m sprint performance at 4, 5, and 6 min recovery intervals after completing five bilateral knee-flexion jumps [56]. This inconsistency may be attributable to the high technical

variability inherent in sprint testing, as participants cannot replicate identical start mechanics and may exhibit postural changes upon completing the sprint [57–59]. Fatigue accumulation could also contribute, as [60] reported that blood lactate concentrations at one minute post-complex potentiation were markedly higher than at ten minutes, and 20m sprint performance improved at ten minutes relative to one minute. This suggests that with increased rest following the intervention, enhanced lactate clearance and reduced neuromuscular fatigue augment sprint performance. As recovery extends further, both potentiation effects and fatigue wane, causing performance to return to baseline. Overall, it is still uncertain when various types of potentiation training most effectively induce the PAPE effect. These outcomes could be affected by various factors such as individual variability, training condition, type of test, and the rate of fatigue recovery [26,61]. Future studies may explore the interplay between various CA types and recovery durations through surface electromyography, offering a more accurate basis for scientifically informed training prescriptions.

## Conclusions

By comparing the acute effects of vertical jump, horizontal jump, and MX PT on short-distance sprint performance, vertical jump demonstrated potential advantages as a pre-competition activation strategy, although these differences were not statistically significant ($p > 0.05$). Given the lack of a control group and the limited sample size, these trend-like findings should still be interpreted with caution. We speculate that the effects of HJ-PT and CJ-PT were diminished, likely due to variations in time-to-peak and cumulative fatigue arising from the complexity of movements. Overall, the results of this study provide preliminary insights into the short-term effects of different potentiation training types on short-distance sprint performance. Future studies should establish no-intervention control conditions while controlling for confounding factors and incorporate surface electromyography to further elucidate the interaction between different CA types and recovery time, thereby providing a more robust foundation for optimizing the PAPE effect.

## Limitations and future prospects

(1)  This study's relatively small sample size and the absence of a control group limits the strength of the conclusions and makes it difficult to separate intervention-induced PAPE from natural variability. Moreover, using a single cohort restricted the exploration of sport-specific PAPE adaptations and reduced generalizability. Future studies should include a control session to address these limitations.

(2) This study lacked robust temperature control to prevent elevated muscle temperature from blunting PAPE. Familiarization and fatigue from four repeated sprints also reduced sprint measure accuracy; future work should assign each time-point to separate interventions and add longer-distance metrics (e.g., 10 m, 20 m) to extend PAPE application.

(3) Future studies should integrate physiological markers—such as electromyography, heart rate, and blood lactate— to elucidate the activation pathways of PAPE, thereby revealing the physiological mechanisms within the neuromuscular system and informing optimization of training strategies.

## Supporting information

**S1 File. Original data.**
(XLSX)

## Acknowledgments

Thank you to all those who have contributed to this work. We thank Dr. Chaoming Liang for his early input on data interpretation and experimental discussions; his contributions are acknowledged here at his own request.

## Author contributions

**Conceptualization:** Wenhao Qu, Duanying Li.

**Data curation:** Wenhao Qu, Wuwen Peng, Yueming Li, Jian Sun, Duanying Li.

**Formal analysis:** Wenhao Qu, Lin Xie, Duanying Li.

**Funding acquisition:** Duanying Li.

**Investigation:** Lin Xie, Duanying Li.

**Methodology:** Jian Sun, Duanying Li.

**Project administration:** Jian Sun, Duanying Li.

**Resources:** Jian Sun, Duanying Li.

**Software:** Duanying Li.

**Supervision:** Jian Sun, Duanying Li.

**Validation:** Duanying Li.

**Visualization:** Duanying Li.

**Writing – original draft:** Wenhao Qu.

**Writing – review & editing:** Wenhao Qu.

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
