## [Decision Letter · Decision Letter 0]

11 Aug 2025

Dear Dr. Li,

Thank you for submitting your manuscript to PLOS ONE. After careful consideration, we feel that it has merit but does not fully meet PLOS ONE’s publication criteria as it currently stands. Therefore, we invite you to submit a revised version of the manuscript that addresses the points raised during the review process.

**ACADEMIC EDITOR:**

We look forward to receiving your revised manuscript.

Kind regards,

Emiliano Cè, Ph.D.

Academic Editor

PLOS ONE

Journal Requirements:

2. Thank you for stating the following financial disclosure: [This work was supported by the Guangdong Provincial Philosophy and Social Sciences Regularization Project 2022 (GD22CTY09): Research on the Coordinated Development Path of International Competitiveness in Sports in the Guangdong-Hong Kong-Macao Greater Bay Area.].

5. If any table files for review show as item type ‘other’ please change to item type ‘table’ as the reviewer does not have access to these ’other’ files.

Reviewers' comments:

Reviewer's Responses to Questions

**Comments to the Author**

1. Is the manuscript technically sound, and do the data support the conclusions?

Reviewer #1: Yes

2. Has the statistical analysis been performed appropriately and rigorously?

Reviewer #1: Yes

3. Have the authors made all data underlying the findings in their manuscript fully available?

Reviewer #1: Yes

4. Is the manuscript presented in an intelligible fashion and written in standard English?

Reviewer #1: Yes

Reviewer #1: General Evaluation

The article addresses a timely and practically relevant topic in sports science — the comparison of the effectiveness of three types of plyometric potentiation training (VJ-PT, HJ-PT, MX-PT) in inducing the post-activation performance enhancement (PAPE) effect in the context of a 5-meter sprint. The study is properly designed and includes detailed analysis, but it also has several limitations that require consideration.

2. Strengths

Appropriately selected study design (randomized crossover)

Use of dynamic warm-up and control of procedural variables

Practical implications for coaches in speed-based sports

Use of Cohen’s d for interpreting effect sizes

Cautious language in the conclusions ("potential effect", "suggested peak")

3. Limitations and Suggestions

1. Lack of a control group:

The study did not include a non-intervention group (placebo or passive control), which limits the ability to determine whether the observed changes were due to the intervention or to natural variability. This omission makes it difficult to conclude whether any form of training works better than “doing nothing.”

Recommendation: Consider addressing this point in the “Limitations” section and, where feasible, include a control group in future studies. This would help clarify whether PAPE is the result of a specific intervention or may arise spontaneously.

2. Lack of information about participants' sport disciplines:

The authors did not specify whether participants were sprinters, football players, or generally active individuals. This is important because:

PAPE is highly dependent on neuromuscular specificity, which differs between, for example, sprinters, football players, gymnasts, and recreational athletes.

The selection of potentiation exercises (VJ-PT, HJ-PT, MX-PT) should be matched to the dominant movement pattern of the athlete’s sport.

The absence of this information limits the applicability of results to specific training contexts.

Recommendation: Please include in the manuscript (e.g., in the “Participants” or “Limitations” section) information about participants’ sport profiles — for example, whether they were students in sport-related programs, speed-based athletes, or individuals engaged in general fitness training.

3. Potential fatigue effects from four sprints in one session:

Performing four sprints (at 4, 8, 12, and 16 minutes post-intervention) may itself induce fatigue, even over such a short distance, potentially impairing subsequent performance. Moreover, it is unclear whether performance changes at 8 minutes reflect PAPE or fatigue from the 4-minute sprint.

A learning or habituation effect may also occur, with participants improving due to test familiarity rather than true potentiation.

Alternative: Consider testing each time point in a separate session to isolate the effects.

4. Inconsistency between assumptions and HJ-PT results:

The authors suggest HJ-PT should enhance sprinting due to horizontal force application, yet the data do not support this. A deeper analysis of this contradiction is warranted.

5. Small sample size (n = 12):

While statistically acceptable as a minimum, this sample size limits the detection of subtle effects and reduces generalizability.

6. No physiological measurements:

The authors attribute MX-PT’s ineffectiveness to fatigue, but no physiological variables (e.g., EMG, HR, lactate) were measured.

Suggestion: Include objective physiological indicators in future research to confirm this interpretation.

7. Potential learning effect:

Repeated sprint testing may improve performance through familiarity rather than PAPE.

Suggestion: Consider randomizing test order or including additional control tests.

8. Use of only 5-meter sprint:

While appropriate for assessing explosive start ability, the 5 m distance does not represent the full acceleration phase.

Suggestion: Future studies could include 10–20 m sprints to assess broader sprint performance.

4. Data Presentation

The data are generally well-presented, but:

There is no graphical display of individual participant results (e.g., spaghetti plots)

Percent change from baseline is not shown

Statistical significance is not marked on graphs (e.g., *, **)

Suggestion: Improve data visualization by including individual trends, percentage changes, and markers for statistical significance.

5. Evaluation of Conclusions

The conclusions are mostly consistent with the data, but:

The lack of statistical significance (p > 0.05) should be explicitly stated

The conclusions regarding HJ-PT are somewhat overstated — effects were minimal or negative

The explanation that MX-PT was ineffective due to fatigue should be presented as a hypothesis, not a fact

6. Final Comment

It is recommended to refine the “Limitations” section, expand the description of participant backgrounds, clarify the interpretation of HJ-PT outcomes, and improve the graphical presentation of results. The study has both scientific and practical value, especially for coaches and researchers focused on PAPE and short-distance sprint performance.

**Do you want your identity to be public for this peer review?** For information about this choice, including consent withdrawal, please see our Privacy Policy

Reviewer #1: No

---

## [Author Response · Author response to Decision Letter 1]

1 Sep 2025

Thank you for your suggestions. The relevant modifications have been made in the document.

---

## [Decision Letter · Decision Letter 1]

8 Oct 2025

Dear Dr. Li,

Thank you for submitting your manuscript to PLOS ONE. After careful consideration, we feel that it has merit but does not fully meet PLOS ONE’s publication criteria as it currently stands. Therefore, we invite you to submit a revised version of the manuscript that addresses the points raised during the review process.

**ACADEMIC EDITOR:**plosone@plos.org . A rebuttal letter that responds to each point raised by the academic editor and reviewer(s). You should upload this letter as a separate file labeled 'Response to Reviewers'.A marked-up copy of your manuscript that highlights changes made to the original version. You should upload this as a separate file labeled 'Revised Manuscript with Track Changes'.An unmarked version of your revised paper without tracked changes. You should upload this as a separate file labeled 'Manuscript'.

We look forward to receiving your revised manuscript.

Kind regards,

Emiliano Cè, Ph.D.

Academic Editor

PLOS ONE

Journal Requirements:

Reviewers' comments:

Reviewer's Responses to Questions

**Comments to the Author**

Reviewer #2: (No Response)

2. Is the manuscript technically sound, and do the data support the conclusions?

Reviewer #2: Partly

3. Has the statistical analysis been performed appropriately and rigorously?

Reviewer #2: Yes

4. Have the authors made all data underlying the findings in their manuscript fully available?

Reviewer #2: Yes

5. Is the manuscript presented in an intelligible fashion and written in standard English?

Reviewer #2: Yes

Reviewer #2: This study aimed to investigate the effects of different plyometric training modalities on post-activation performance enhancement (PAPE) in short-distance sprint performance. This topic can give practical applications to coaches that work in a sport where short sprint and acceleration are relevant for performance.

Below, I provide my comments, first focusing on general aspects and then on specific points that may help the authors improve the quality of their work.

GENERAL ASPECTS:

1. The absence of a control group limits the strength of the conclusions that can be drawn from this study. Given that no significant interactions were found, including a control session would be useful mainly for completeness of the study design. However, since this was not done, I recommend clearly emphasizing this limitation in the Conclusions section, as it substantially affects the interpretation of the results.

2. In the second part of the “INTRODUCTION” (Line 70-92) I would suggest the authors be more precise and explicit when describing the concept of conditioning activity (CA) in the context of PAPE. Furthermore, the term Potentiation Training requires clarification: please specify what is meant, and, if it has been previously used in the literature, provide the appropriate reference(s).

3. In the Statistical Analysis section, I would suggest revising the part mentioning the Bonferroni post hoc test. Since no significant interactions were found, the post hoc reference may be unnecessary or should be presented as hypothetical. Therefore, the sentence “Bonferroni post hoc...significant” could be removed for clarity. Additionally, I would strongly recommend reporting the “ANOVA effect size” instead of the “between-group effect size”, and removing the expression “for non-significant comparisons.” (line 164)

4. Since the main results are not statistically significant, I recommend revising the Discussion and Conclusions accordingly. Given the absence of significant conditions and the lack of a control group, even the reported medium effect size might be overestimated. Therefore, I strongly recommend rewriting the Discussion in light of the non-significant findings to ensure that the interpretation remains balanced and consistent with the data.

5. Throughout the manuscript (i.e. line 173), the use of the term “group” appears to be misleading, as it may imply the presence of distinct participant groups. I strongly recommend replacing it with “intervention” or “condition”, which would more accurately reflect the study design and avoid potential confusion for the reader.

SECIFIC POINTS

Title (line 1-5): I suggest the authors using the world “plyometric” to characterize the potentiation training. Instead of “Different Forms of Potentiation Training” you could evaluate the use of “Plyometric Potentiation Training” or “Plyometric Training Modalities”.

Line 12: Please, add some generale information (sex, age, BMI) of the participant in a bracket after “12 participants”

Line 31: After “2 sets x 6 repetitions”, please add “of one of the jumps training modalities” and specify the rest time between sets.

Line 32: I suggest using “pre intervention” instead of “at baseline”, but consider keeping it like it is.

Line 53: I suggest the authors to use only the words voluntary and involuntary (or induced).

Line 55-57 : Please consider re-writing the sentence, the use of “excited” and “endogenous” is awkward.

Line 57: : I strongly suggest the authors to the sentence after “rate of force development”, delate “or explosive strength of the muscle” because is too general.

Line 58-59: Instead of “in the neuromuscular system following a single intense stimulation”, please consider re-writing the sentence “induced by a single intense stimulation”.

Line 60: instead of “enhances cross-bridges cycle efficiency”, use “increase the rate of cross-bridge formation”.

Line 65: please separate “10” with “minutes”

Line 70: please separate “(12)” and “to”; after “athletic performance” add “after a conditioning activity (CA)”.

Line 103: after each training intervention write in brackets the acronym

Line 109: please separate “72” and “hours”

Line 113: Regarding Figure 1, I would recommend making it more essential and easier to read. Since you have already stated that the protocols were randomized, I would suggest avoiding repetition in the box below. Instead of writing '6 × 2' it would be clearer to specify '2 sets of 6 reps.' In general, try to minimize the amount of text included (for instance, the note under Session 1 could be omitted or made more concise).

Line 117: I suggest the author changing with “was determined to be 12 participants.” Than start the sentence with “Based on …”

Line 119: Instead of “established. (1) No history…” please use “established: (1) no history…”

Line 127: I could not find the table 1 referred to in the manuscript. Could the authors please clarify whether it corresponds to the one included in the original dataset? In addition, please ensure that the table provides all the necessary information about the participants, including sample size, sex, age, BMI, and any other relevant characteristics.

Line 129: It is not clear from this sentence which measurements were taken and with which instruments. Please clarify.

Lines 131–133: It is not clear whether the warm-up consisted only of the specific exercises listed, or if it also included a general warm-up (e.g., cycle ergometer, running). If a general warm-up was performed, please specify which activity was used.

Line 133-137: Please revise the phrasing of these sentences. The full stop after 'pre-test (35)' appears unnecessary and should be removed.

Line 139: I could not find Table 2 in the manuscript. Please check and ensure it is properly included.

Line 172: I could not find Table 3 in the manuscript. Please check and ensure it is properly included.

Line 176: I would suggest using 'pre-intervention' instead of 'baseline' to improve clarity.

Line 178: I could not find Table 3 in the manuscript. Please check and ensure it is properly included.

Line 174-176: I recommend removing “Post-hoc comparisons … in magnitude”.

Figure 2: The figure lacks the unit of measurement on the Y-axis and does not clearly specify which variable is being represented. Overall, the figure is not easily interpretable and appears redundant, as it closely resembles Figure 3, which is more comprehensive. I would suggest considering its removal to improve clarity and conciseness.

Figura 3: I would encourage the authors to focus on improving this figure. It would be helpful to highlight individual subjects using a different and more transparent symbol, and then overlay the group mean trend with a distinct symbol and a more prominent line. You might also consider displaying the individual data in black and white, while using a different color for the mean. Please also ensure consistency in the labeling of the X-axis — for example, the “min” label should have the same font style across all time points.

Given that the results are not statistically significant, I would recommend removing Figures 4 and 6, as they do not add meaningful information to the manuscript.

**Do you want your identity to be public for this peer review?** For information about this choice, including consent withdrawal, please see our Privacy Policy

Reviewer #2: No

---

## [Author Response · Author response to Decision Letter 2]

19 Oct 2025

We sincerely thank you for your meticulous review of our manuscript. Your exhaustive suggestions on English wording and grammar have made us deeply aware of the considerable time and effort you devoted to this process, and we hold the utmost respect for your work. In response to your guidance, we have addressed every comment and detailed all revisions in the attached file “Response to Reviewer.” Should any further issues arise, we will promptly revise the manuscript accordingly.

---

## [Decision Letter · Decision Letter 2]

11 Dec 2025

Dear Dr. Li,

Thank you for submitting your manuscript to PLOS ONE. After careful consideration, we feel that it has merit but does not fully meet PLOS ONE’s publication criteria as it currently stands. Therefore, we invite you to submit a revised version of the manuscript that addresses the points raised during the review process.

**ACADEMIC EDITOR:**plosone@plos.org . A rebuttal letter that responds to each point raised by the academic editor and reviewer(s). You should upload this letter as a separate file labeled 'Response to Reviewers'.A marked-up copy of your manuscript that highlights changes made to the original version. You should upload this as a separate file labeled 'Revised Manuscript with Track Changes'.An unmarked version of your revised paper without tracked changes. You should upload this as a separate file labeled 'Manuscript'.

We look forward to receiving your revised manuscript.

Kind regards,

Emiliano Cè, Ph.D.

Academic Editor

PLOS One

Journal Requirements:

Reviewers' comments:

Reviewer's Responses to Questions

**Comments to the Author**

Reviewer #2: (No Response)

2. Is the manuscript technically sound, and do the data support the conclusions?

Reviewer #2: Yes

3. Has the statistical analysis been performed appropriately and rigorously?

Reviewer #2: Yes

4. Have the authors made all data underlying the findings in their manuscript fully available?

Reviewer #2: Yes

5. Is the manuscript presented in an intelligible fashion and written in standard English?

Reviewer #2: Yes

Reviewer #2: This study aimed to investigate the effects of different plyometric training modalities on post-activation performance enhancement (PAPE) in short-distance sprint performance. This topic can give practical applications to coaches that work in a sport where short sprint and acceleration are relevant for performance.

I appreciate the effort you have made to revise the manuscript according to the previous feedback. The paper is now clearer and more structured. Nonetheless, a few important issues remain that should be addressed to further enhance the quality and clarity of your work.

GENERAL ASPECT

Please revise the manuscript carefully with particular attention to spacing and punctuation. In several sections, spacing is inconsistent (e.g., missing spaces after commas or before parentheses), which affects the overall readability and professionalism of the paper. A thorough formal review of the text is recommended to ensure consistency throughout.

SPECIFIC POINTS

Title (line 1-5): I suggest the authors evaluate the use of the world “different” before “Pliometric Training Modalities”. The title should sound like “The Post-Activation Performance Enhancement Effect of Different Plyometric Training Modalities on Short-Distance Sprinting: An Acute Randomized Crossover Study”

Line 27: I suggest using “CJ -PT” instead of “MX-PT”, to identify “combined jump plyometric training”.

Line 41: I suggest using “short-term effects of different plyometric based conditioning activities” instead of “short-term effects of conditioning activities”.

Line 27: I suggest using “CJ -PT” instead of “MX-PT”, to identify “combined jump plyometric training”. So use also the world “combined” instead of “mixed”.

Line 50: Instead of “excitation”, please consider re-writing the sentence using “potentiation”.

Line 53-54: I recommend revising this sentence for clarity and readability. Instead of “the neuromuscular system is activated endogenously through maximal or near-maximal voluntary contractions, resulting in an acute increase in the rate of force development,” you may consider using the following formulation:

“…the neuromuscular system performance is enhanced through maximal or near-maximal voluntary contractions, resulting in an acute increase in force production and in the rate of force development.”

Line 57-58: To improve clarity, you might consider rephrasing it as follows : “Over the years, this principle has become a classic component of sports training and has been increasingly translated into practical applications within sport performance settings.”

Line 97-98: This sentence could be improved for clarity and readability, I suggest “All participants completed two familiarization sessions to ensure they were fully comfortable with the sprint tests and the various plyometric exercises before the experimental sessions.”

Line 104: please consider adding “and fatigue” after “residual training”.

Line 112: I suggest using “:” after “established”

In the paragraph “Post-activation performance enhancement protocols”, please clarify the structure of the combined jump plyometric training (which I recommend referring to as CJ-PT for consistency). Specifically, it is not entirely clear whether one repetition consists of a tuck jump immediately followed by a horizontal frog jump. If this is the case, please state it explicitly to ensure the protocol is fully understandable and replicable. Additionally, you may consider revising the title of this paragraph to better reflect its content. For example, “Plyometric Conditioning Activity Protocols” might provide a clearer and more accurate description.

Line 151: Please define the acronym ANOVA the first time it appears in the manuscript. Even though it is a well-known statistical term, it is good practice to write it out in full before introducing the abbreviation. In this case, you may use: “Analysis of Variance (ANOVA)”.

Line 152-153: I recommend revising this sentence for clarity. Instead of “Between-intervention effect sizes (ES) were reported as ηp², and within-intervention ES as Cohen’s d (41)”, you may consider using the following formulation:

“The ANOVA effect sizes (ES) were reported as ηp², and for pairwise comparisons Cohen’s d (41) was reported.”

Regarding Figure 1, I have a few suggestions to improve clarity and consistency:

1. Please align the "SESSION 1" box with the other session boxes for visual consistency. If "SESSION 1" refers to the two familiarization sessions, I recommend specifying them as “FAMILIARIZATION 1” and “FAMILIARIZATION 2” to avoid confusion.

2. In the PAPE box, I suggest adding a space for readability: “PAPE 2 sets of 6 reps (Self respect)” instead of “PAPE 2 sets of 6 reps(Self respect)”.

3. Please clarify the meaning of “Self respect” in this context, as it is unclear to the reader.

In Table 1, please adjust the formatting by placing the closing parenthesis immediately after “years” to ensure consistency and correct punctuation.

In Table 2, please clarify the meaning of “self respect”, as its interpretation within the context of the protocol is not evident. Additionally, the term “intergroup interval” requires clarification: does it refer to the recovery between repetitions or between sets? Please specify this to avoid ambiguity. In the first column, I suggest replacing the term “Group” with “CA”, “Conditioning Activity”, or “Plyometric CA”, as these alternatives provide a more accurate and descriptive label for the conditioning stimulus used. I also remind you to consider replacing MX-PT with CJ-PT, as previously recommended, to ensure consistent terminology throughout the manuscript.

In Figure 2, please include the unit of measurement on the y-axis. In this case, time (s) should be specified to ensure clarity and consistency with standard reporting practices.

In Figure 2 and Table 3, please remember to consider replacing MX-PT with the revised terminology previously suggested (e.g., CJ-PT), to maintain consistency throughout the manuscript.

In Figure 3, please ensure consistency in the symbols used to represent the horizontal vs. mixed (which I recommend renaming to combined) conditions. At 4, 8, and 16 minutes you used a diamond symbol, whereas at 12 minutes a circle appears. This inconsistency may confuse readers, so I recommend standardizing the symbol across all time points.

**Do you want your identity to be public for this peer review?** For information about this choice, including consent withdrawal, please see our Privacy Policy

Reviewer #2: No

---

## [Author Response · Author response to Decision Letter 3]

16 Dec 2025

We sincerely thank the reviewer for the thorough evaluation of our manuscript and for the constructive and insightful comments, as well as for the considerable time and effort devoted to improving the quality of our work. We are especially grateful for the generous and selfless guidance regarding our English writing. As English is not our first language, unintentional issues and oversights inevitably occurred during the manuscript preparation process. We truly appreciate the reviewer’s patience, understanding, and thoughtful suggestions, which are of great value to us and will serve as strong motivation for our future academic work, extending well beyond the present manuscript.

---

## [Decision Letter · Decision Letter 3]

9 Jan 2026

Dear Dr. Li,

Thank you for submitting your manuscript to PLOS ONE. After careful consideration, we feel that it has merit but does not fully meet PLOS ONE’s publication criteria as it currently stands. Therefore, we invite you to submit a revised version of the manuscript that addresses the points raised during the review process.

**ACADEMIC EDITOR:**plosone@plos.org . A letter that responds to each point raised by the academic editor and reviewer(s). You should upload this letter as a separate file labeled 'Response to Reviewers'.A marked-up copy of your manuscript that highlights changes made to the original version. You should upload this as a separate file labeled 'Revised Manuscript with Track Changes'.An unmarked version of your revised paper without tracked changes. You should upload this as a separate file labeled 'Manuscript'.

We look forward to receiving your revised manuscript.

Kind regards,

Emiliano Cè, Ph.D.

Academic Editor

PLOS One

Journal Requirements:

Reviewers' comments:

Reviewer's Responses to Questions

**Comments to the Author**

Reviewer #2: All comments have been addressed

2. Is the manuscript technically sound, and do the data support the conclusions?

Reviewer #2: Yes

3. Has the statistical analysis been performed appropriately and rigorously?

Reviewer #2: Yes

4. Have the authors made all data underlying the findings in their manuscript fully available?

Reviewer #2: Yes

5. Is the manuscript presented in an intelligible fashion and written in standard English?

Reviewer #2: Yes

Reviewer #2: This study aimed to investigate the effects of different plyometric training modalities on post-activation performance enhancement (PAPE) in short-distance sprint performance. This topic can give practical applications to coaches that work in a sport where short sprint and acceleration are relevant for performance.

I appreciate the effort you have made to revise the manuscript according to the previous feedback. The paper is now clearer and more structured. However, a few minor aspects can still be improved, particularly regarding the graphical presentation of the figures and the formal presentation of the text, as outlined below.

SPECIFIC POINTS

At line 33, in the sentence “5-meter sprint performance pre-intervention and at 4,8,12and16 minutes post-training”, there is a lack of spacing between the time points. I recommend revising it for correct formatting and readability. For example, it could be rewritten as:

“5-meter sprint performance pre-intervention and at 4, 8, 12, and 16 minutes post-training.”

In the Conclusion of the Abstract, you state that “Future studies should control confounding factors and use surface electromyography to clarify interactions between CA types and recovery.”

Given that one of the main limitations of the present study is the sample size, I recommend explicitly adding that future studies should also include a larger sample and further investigate the responses in both sexes.

At line 85, please insert a space between “mechanism” and the reference in parentheses to ensure correct formatting.

At line 115, please insert a space between “72” and “hours” to ensure correct formatting.

At line 137, please insert a space between “10” and “minutes” to ensure correct formatting.

At line 206, please insert a space between “effect” and the reference in parentheses to ensure correct formatting.

At line 209, please insert a space between “acceleration” and the reference in parentheses to ensure correct formatting.

At line 216, please insert a space between “performance” and the reference in parentheses to ensure correct formatting.

At line 218, please insert a space between “100ms” and the reference in parentheses to ensure correct formatting.

Please also verify whether similar spacing issues occur in other parts of the manuscript and correct them accordingly.

At line 242, there is an extra opening parenthesis. Please remove it to ensure correct punctuation and formatting.

At line 251, please insert a period before “Future” to correctly separate the sentences.

Re-reading the manuscript, I noticed an inconsistency between the Study Design section and Figure 1. Specifically, at lines 113–117 you state that the intervention protocols were separated by a minimum of 72 hours, whereas Figure 1 indicates a 48-hour interval between sessions. Please clarify this discrepancy and ensure consistency between the text and the figure.

Regarding Figure 1:

1. If “SESSION 1” refers to the two familiarization sessions conducted on different days, as I understand, I recommend removing “Session 1” and retaining only “Familiarization 1” and “Familiarization 2” as two separate boxes. The remaining boxes should then be renumbered accordingly.

2. In the PAPE section, I suggest using VJ-PT, HJ-PT, and CJ-PT, respectively, instead of “Process 1”, “Process 2”, and “Process 3”, to improve clarity and consistency with the terminology used in the manuscript.

Regarding Figure 3:

Please ensure consistency in the spacing between the time points and the box. Currently, the first two time points (16 min and 12 min) are placed too close to the box compared with the other two (8 min and 4 min), which affects the visual balance of the figure.

In Table 2 and Figure 2, please note that the unit seconds should be reported using a lowercase “s” for consistency with SI units.

**Do you want your identity to be public for this peer review?** For information about this choice, including consent withdrawal, please see our Privacy Policy

Reviewer #2: No

---

## [Author Response · Author response to Decision Letter 4]

13 Jan 2026

We sincerely thank Reviewer #2 for the positive evaluation of our manuscript and for the constructive comments, which have helped us further improve the clarity, consistency, and presentation quality of the paper. We have carefully addressed all the points raised. Our detailed responses are provided below.

---

## [Decision Letter · Decision Letter 4]

26 Jan 2026

The Post-Activation Performance Enhancement Effect of Different Plyometric Training Modalities on Short-Distance Sprinting: An Acute Randomized Crossover Study

PONE-D-25-30573R4

Dear Dr. Li,

We’re pleased to inform you that your manuscript has been judged scientifically suitable for publication and will be formally accepted for publication once it meets all outstanding technical requirements.

Kind regards,

Emiliano Cè, Ph.D.

Academic Editor

PLOS One

Additional Editor Comments (optional):

Reviewers' comments:

Reviewer's Responses to Questions

**Comments to the Author**

Reviewer #2: All comments have been addressed

2. Is the manuscript technically sound, and do the data support the conclusions?

Reviewer #2: Yes

3. Has the statistical analysis been performed appropriately and rigorously?

Reviewer #2: Yes

4. Have the authors made all data underlying the findings in their manuscript fully available?

Reviewer #2: Yes

5. Is the manuscript presented in an intelligible fashion and written in standard English?

Reviewer #2: Yes

Reviewer #2: In my opinion, the authors have substantially improved the manuscript, and the revisions have significantly increased the overall clarity and quality of the article. The study now reads much more coherently and the methodological aspects are better justified.

I would like to once again draw the authors’ attention to point 6 raised in my previous review. Specifically, I kindly ask the authors to verify that in Figure 1 the time interval between sessions is reported as 72 h instead of 48 h, in accordance with what is stated in the main text of the manuscript.

**Do you want your identity to be public for this peer review?** For information about this choice, including consent withdrawal, please see our Privacy Policy

Reviewer #2: No

---

## [Editor Report · Acceptance letter]

PONE-D-25-30573R4

PLOS One

Dear Dr. Li,

I'm pleased to inform you that your manuscript has been deemed suitable for publication in PLOS One. Congratulations! Your manuscript is now being handed over to our production team.

Kind regards,

on behalf of

Prof. Emiliano Cè

Academic Editor

PLOS One